# Effects of Different Aging Methods on the Phenolic Compounds and Antioxidant Activity of Red Wine

**Chao Wang** [1,2], **Chenhui Wang** [1,2], **Ke Tang** [1,2], **Zhiming Rao** [1,2] **and Jian Chen** [1,2,*]

1   Key Laboratory of Industrial Biotechnology, Ministry of Education, School of Biotechnology, Jiangnan University, Wuxi 214122, China
2   Science Center for Future Foods, Jiangnan University, Wuxi 214122, China
*   Correspondence: jchen@jiangnan.edu.cn

**Abstract:** In this study, oak barrels, glazed pottery altars, unglazed pottery altars, and stainless-steel tanks were selected as aging containers for red wine, and phenolic compounds and antioxidant activity were analyzed and compared. The color of red wine in unglazed pottery altars and glazed pottery altars changed to brick red and brownish yellow, respectively; the color of red wine in oak barrels did not change significantly; and color retention was best in stainless-steel tanks. The total content of anthocyanins and nonanthocyanin phenolic compounds was higher in the unglazed pottery altar group (227.68 mg/L and 288.88 mg/L, respectively) than in the oak barrel group (209.46 mg/L and 273.42 mg/L), the stainless-steel tank group (221.92 mg/L and 213.23 mg/L), or the glazed pottery altar group (74.71 mg/L and 204.43 mg/L). After aging, DPPH (1,1-diphenyl-2-picrylhydrazine free radical scavenging ability), I confirm. (ABTS$^+$ free radical scavenging ability), and FRAP (a ferric ion-reducing antioxidant power reduction of Ion Ability) were decreased by 8.8%, 0.5%, and 17.1%, respectively, in the unglazed pottery altar group; by 15.2%, 1.7%, and 19.5%, respectively, in the oak barrel group; by 18.0%, 1.8%, and 20.0%, respectively, in the stainless-steel tank group; and by 18.7%, 4.2%, and 34.9%, respectively, in the glazed pottery altar group. In conclusion, antioxidative ability decreased less in the unglazed pottery altar group than in the other three groups, indicating that unglazed pottery altars retain antioxidant components in red wine well.

**Keywords:** red wine; color; phenolic compounds; antioxidant; aging method

## 1. Introduction

Polyphenols have always been a hotspot in wine research in view of their important roles in red wine [1–3]. Studies have shown that polyphenols have antioxidant, anti-aging, anti-inflammatory, antibacterial, anti-tumor, and arteriosclerosis-preventive effects [4–7]. In addition, polyphenols play a very important role in color and taste of red wine itself [8]. Polyphenol levels in red wine and interactions between polyphenols affect color and stability of red wine [9]. Colorless polyphenols such as catechins can interact with anthocyanins and play roles as co-colors. Jacinto et al. showed that co-pigments in wine aged for 2 years contributed 18.5% to wine color [10].

Many factors affect polyphenol content in wine. In addition to grape variety, aging container is also a key factor. Oak barrels, the most widely used wine storage containers in the world, have unique advantages. Oak contains polyphenols such as vanillin, eugenol, and guaiacol, as well as their derivatives, all of which are slowly absorbed by wine during the aging process, making that wine more mellow and full. In addition, oak barrels are toasted during the production process, which makes the aroma of wine more complex by giving it vanilla, almond, coffee, cream, caramel, cigar, chocolate, toast or smoked bacon aroma, or other roasted aromas [11]. Tiny air holes on the surface of oak barrels also create a unique "micro-oxygen environment" in which wine can be properly oxidized, promoting its maturity, softening tannins in wine, and improving taste of wine [12]. However, oak

barrels currently face many problems. First, oak resources are scarce and utilization rate is low. Second, the price of oak barrels is high. The price of a 225-L oak barrel made of French oak is about 600 USD. The investment in oak barrels for aging 1 ton of wine is 2400 USD, and this price is still increasing. Third, repeated use of oak barrels leads to a sharp decline in beneficial substances, so oak barrels generally last only 2–3 years [13].

Therefore, it is necessary to explore and develop new aging vessels. In our survey of other aging vessels, we noticed that the pottery altar, one of the oldest storage vessels, was rarely mentioned. Rossetti et al. compared the tastes and aromas of wines stored in amphorae and barrels. After 6 months of aging, wines made in amphorae had a spicy taste, contained pleasant tannins, and had a less "green" character than barrel-aged wines, but also contained weak aromas [14]. Baiano et al. stored white wine in glass containers and three types of amphorae. After 12 months of aging, contents of flavonoids and vanillin-reactive flavans were significantly reduced in wine samples from raw amphorae. Wine in Engobe amphorae had the highest antioxidant activity, while wine in glass containers and glazed amphorae had the lowest antioxidant activity [15].

However, it is not yet clear how aging in pottery altars affects wine color, phenolic content, or antioxidant activity. In the present study, oak barrels, glazed pottery altars, unglazed pottery altars, and stainless-steel tanks were selected as aging containers for red wine. Color and phenolic content of red wine were analyzed after aging. The effects of pottery altars on red wine were analyzed and compared with those of two other vessels. In this way, we explored the advantages of pottery altars as aging containers and examined the possibility of replacing oak barrels.

## 2. Materials and Methods

### 2.1. Chemicals

Catechin hydrate (≥98%, HPLC), salicylic acid (≥99%, HPLC), morin hydrate (≥95%, HPLC), rutin hydrate (≥95%, HPLC), gallic acid (≥98%, HPLC), protocatechuic acid (≥97%, HPLC), *p*-hydroxybenzoic acid (≥97%, HPLC), vanillic acid (≥97%, HPLC), syringic acid (≥95%, HPLC), epicatechin (≥98%, HPLC), gentisic acid (≥98%, HPLC), caffeic acid (≥98%, HPLC), epicatechin gallate (≥98%, HPLC), ellagic acid (≥95%, HPLC), quercetin (≥95%, HPLC), and anthocyanin-3-*O*-glucoside (≥95%, HPLC) were obtained from Sigma-Aldrich (St. Louis, MO, USA). Methanol (chromatographically pure) was obtained from the Shanghai Anpu Scientific Instrument Company. Ethanol (chromatographically pure) was obtained from Sigma-Aldrich (St. Louis, MO, USA). Acetonitrile (chromatographically pure) was obtained from the CNW Company. Folin-Ciocalteu was obtained from the Shanghai Macklin Company. DPPH, ABTS, and FRAP were obtained from the Shanghai Yuanye Company (Shanghai, China).

### 2.2. Samples

Red wines were placed in 5 L French oak barrels (*Quercus petraea*) (moderately roasted, French Group Viard Company, East Sussex, France, Country), 5 L unglazed pottery altars (Qufu Hengtong Wine Container Company, Qufu, China), 5 L glazed pottery altars (Qufu Hengtong Wine Container Company, Qufu, China), and 5 L stainless-steel tanks (Qufu Hengtong Wine Container Company, Qufu, China), then aged at 20 °C. Samples were taken for analysis on the 270th day of aging. Trials were repeated three times for each aging vessel. For specific sample information, see Table 1.

### 2.3. Color Analysis

Using distilled water as a control, absorbance values of samples were measured at 440, 530, and 600 nm, and the CIELAB method [16,17] was used to calculate brightness L*, red hue (a*), yellow hue (b*), chromaticity (C*), and hue (h*).

The three parameters of lightness (L*), red hue (a*), and yellow hue (b*) were matched to corresponding color points on the CIELAB 3D axis. Color change was determined based

on a* and b*, and color-level change was determined based on L*. The CIELAB parameter calculation results were color-matched using Adobe Photoshop.

**Table 1.** Red wine samples.

| Sample | Variety | Region | Aging Vessel | Sampling Time |
|--------|---------|--------|--------------|---------------|
| Day0 | Cabernet Sauvignon | Ningxia | None | Day 0 |
| O-270 | Cabernet Sauvignon | Ningxia | Oak barrel | Day 270 |
| T-270 | Cabernet Sauvignon | Ningxia | Unglazed pottery altar | Day 270 |
| Y-270 | Cabernet Sauvignon | Ningxia | Glazed pottery altar | Day 270 |
| G-270 | Cabernet Sauvignon | Ningxia | Stainless-steel tank | Day 270 |

*2.4. Detection of Anthocyanins in Red Wine*

Anthocyanins were detected as previously described [18], with slight modifications. A Waters ultra-high-performance tandem liquid chromatography–triple quadrupole mass spectrometry system was used to detect anthocyanins. A BEH C18 chromatographic column (i.d. 100 mm × 2.1 mm, 1.7 μm, Waters) was used. Column temperature was 45 °C. Injection volume was 5 μL. Flow rate was 0.3 mL/min. Detection wavelength was 520 nm. Mobile phase A was pure acetonitrile and mobile phase B was 2% formic acid (FA). Elution gradient was as follows: 0–20 min, 2–16% A; 20–28 min, 16–23% A; 28–30 min, 23–50% A; 30–35 min, 50–100% A; 35–37 min, 100% A; 37–38 min, 100–2.0% A. The mass spectrometer used electrospray ionization (ESI) and was operated in positive ion mode. Ion scanning range was 100–1000 *m/z*, nebulizer pressure was 35 psi, drying gas flow rate was 10 L/min, and drying gas temperature was 325 °C. Red wine samples were filtered through a 0.45 μm filter and measurements were immediately performed. The anthocyanin standard (Malvidin-3-O-glucoside, ≥95%, HPLC) was dissolved and diluted to different concentrations. After filtration through a 0.22-μm filter membrane, liquid chromatography analysis was performed. With concentration as an abscissa and liquid chromatography peak area as an ordinate, a standard curve equation was obtained as y = 10.516x + 11.651 and the correlation coefficient $R^2$ was 0.9979, meeting the quantitative analysis standard for anthocyanins.

*2.5. Detection of Nonanthocyanin Phenolic Compounds in Red Wine*

A total of 15 nonanthocyanin phenolic compounds in red wine were detected in all samples by liquid chromatography–triple quadrupole mass spectrometry and triple quadrupole compound linear ion trap liquid mass spectrometry, as previously reported [18], with slight modifications. A BEH C18 column (i.d. 100 mm × 2.1 mm, 1.7 μm, Waters) was used. Column temperature was 40 °C, and 2 μL sample was used per injection. Mobile phase A was water containing 0.1% FA and mobile phase B was pure acetonitrile. Flow rate was 0.3 mL/min. Elution gradient was as follows: 0–1 min, 98% A; 1–6 min, 2–98% A; 6–8 min, 2% A; 8–8.1 min, 2–98% A; 8.1–10 min, 98% A. The mass spectrometer used ESI and was operated in negative ion mode. Ion scanning range was 50–1200 *m/z*, nebulizer pressure was 20 psi, drying gas temperature was 325 °C, and drying gas flow rate was 10.00 L/min. After filtration through a 0.22 μm filter, measurements were immediately performed. According to qualitative results, nonanthocyanins contained in the sample were determined, and standard curves were drawn by configuring the standard with the appropriate concentration. Details are shown in Table 2.

**Table 2.** Standard curves of nonanthocyanidin phenolic compounds.

| Number | Monomer Phenolic Compounds | Remaining Time (min) | Standard Curves | $R^2$ | Linear Range $(mg \cdot L^{-1})$ |
|---|---|---|---|---|---|
| 1 | CAT | 2.89 | $y = 1.96365 \times 10^5 x + 2.80011 \times 10^5$ | 0.99531 | 7.62–131.12 |
| 2 | Salicylic acid | 3.78 | $y = 2.88810 \times 10^6 x + 1.05181 \times 10^7$ | 0.99673 | 4.51–17.79 |
| 3 | Rutin | 3.27 | $y = 1.99758 \times 10^6 x - 5.90991 \times 10^5$ | 0.99759 | 0.35–1.46 |
| 4 | Gallic acid | 1.28 | $y = 1.92804 \times 10^6 x - 9.96384 \times 10^5$ | 0.99592 | 6.46–206.84 |
| 5 | Protocatechuic acid | 2.43 | $y = 3.69943 \times 10^6 x - 4.48883 \times 10^6$ | 0.99358 | 0.97–4.03 |
| 6 | 4-Hydroxybenzoic acid | 2.84 | $y = 2.97624 \times 10^6 x + 2.03029 \times 10^6$ | 0.99026 | 0.57–3.98 |
| 7 | Vanillic acid | 3.06 | $y = 2.70277 \times 10^6 x - 1.95991 \times 10^5$ | 0.99897 | 3.07–24.82 |
| 8 | Syringic acid | 3.11 | $y = 8.73529 \times 10^4 x - 3.78496 \times 10^4$ | 0.99510 | 3.01–61.43 |
| 9 | EC | 3.07 | $y = 5.13096 \times 10^5 x - 2.57807 \times 10^5$ | 0.99568 | 1.51–58.94 |
| 10 | EGC | 2.76 | $y = 1.44898 \times 10^6 x - 1.46819 \times 10^6$ | 0.99912 | 1.23–9.92 |
| 11 | 2,5-Dihydroxybenzoic acid | 2.86 | $y = 6.86815 \times 10^6 x - 5.07039 \times 10^6$ | 0.99911 | 0.99–3.66 |
| 12 | Caffeic acid | 3.03 | $y = 2.17934 \times 10^5 x - 1.52866 \times 10^5$ | 0.99777 | 2.44–39.49 |
| 13 | EGCG | 3.07 | $y = 1.44283 \times 10^6 x - 1.14893 \times 10^6$ | 0.99937 | 0.48–3.78 |

The abbreviations and full names of some compounds in Table 2 are as follows: Catechin hydrate (CAT), L-Epicatechin (EC), Epigallocatechin (EGC), Epigallocatechin gallate (EGCG).

*2.6. Determination of DPPH Free Radical Scavenging Ability*

The samples' 1,1-diphenyl-2-picrylhydrazine (DPPH) scavenging activity was determined as previously described [19], with minor modifications. First, 12.5 mg DPPH was dissolved in methanol. The volume was made up to 100 mL, and then the sample was diluted 5 times to obtain 2.00 mM DPPH solution in methanol. Next, 0.10 mL red wine sample was added to 3.90 mL DPPH solution in methanol, and samples were mixed well. The reaction was allowed to take place in the dark for 30 min, and absorbance was measured at 516 nm. As a blank control, 10.0% methanol was used. Results were expressed as DPPH (%). DPPH scavenging rate was calculated as follows:

$$DPPH(\%) = \frac{Ai - Aj}{Ai} \times 100\% \qquad (1)$$

where Ai represents absorbance of the blank control at 516 nm and Aj represents absorbance of the red wine sample at 515 nm.

*2.7. Determination of ABTS Free Radical Scavenging Ability*

Since antioxidants in wine can combine with oxidants to convert green $ABTS^+$ into colorless ABTS, total antioxidant capacity of samples can be calculated by measuring absorbance value of ABTS free scavenging ability. Total ABTS scavenging activity was determined with a commercial kit, as previously reported [20,21], with minor modifications. Specific steps were as follows: Hydrogen peroxide solution was diluted 1000 times with double-distilled water. An appropriate amount of peroxidase was taken and diluted 10 times with the detection buffer. ABTS working solution was prepared by mixing 152 µL detection buffer with 10 µL ABTS working solution and 8 µL 1:1000 hydrogen peroxide solution. Standard solution was diluted with distilled water, and 20 µL peroxidase working solution per well was added to a 96-well plate. In addition, 10 µL distilled water was added to blank control wells, 10 µL Trolox standard solution of various concentrations was added to standard-curve wells, and 10 µL wine sample was added to sample-detection wells. Finally, 170 µL ABTS working solution was added to each well, and samples were mixed gently and incubated at room temperature for 6 min. Absorbance was measured at 414 nm ($A_{414}$).

The blank $A_{414}$ value was subtracted from each sample value. The following Trolox standard curve was drawn: $y = 0.8846x + 0.0641$, $R^2 = 0.9973$. Total antioxidant activity of each sample was calculated using this formula.

### 2.8. Determination of FRAP Reduction of Ion Ability

To test the antioxidant capacity of wine samples, a ferric ion-reducing antioxidant power (FRAP) assay was performed with a commercial kit, as previously reported [22]. FRAP working solution was prepared by mixing 150 μL tripyridyltriazine (TPTZ) dilution solution with 15 μL TPTZ solution and 15 μL detection buffer. After distributing 180 μL FRAP working solution to each well of a 96-well plate, 5 μL sample was added to the testing well and 5 μL distilled water was added to the blank control well. Next, the plate was incubated at 37 °C for 3–5 min and $A_{593}$ was measured. A standard curve was prepared by using FeSO4 solution at different concentrations, described by the following typical equation: $y = 0.3x - 0.2431$ ($R^2 = 0.9982$). The total antioxidant activity was calculated according to this equation.

### 2.9. Statistical Analysis

All sample measurements were repeated three times, and statistical analysis was performed using GraphPad Prism 8.0.2. A one-way analysis of variance (ANOVA) followed by Bonferroni was performed using SigmaStat version 4.0 for Windows (SYSTAT Software Inc., San Jose, CA, USA). Origin 2021 was used for drawing.

## 3. Results and Discussion

### 3.1. Determination of Color Parameters of Red Wine

The color parameters of red wine samples stored in different containers are shown in Table 3. Yellow hue of red wine stored in glazed pottery altars was higher than that of wine stored in other containers. Color lightness (L*) of wines in the four aging vessels decreased, meaning that color changed from dark to light. Change in brightness was larger for wines aged in oak barrels and in glazed pottery altars than for wines aged in the other two aging vessels, with the least change for wines aged in stainless-steel tanks. Red hue (a*) varied less overall, with oak-barrel- and stainless-steel-tank-aged red wines showing an increasing trend while other wines showed a decrease. The most obvious change was observed for yellow hue (b*), which gradually increased in wines in all four vessels, with the greatest change observed in wine aged in glazed pottery altars.

**Table 3.** CIELAB color parameters of red wine samples in different aging containers.

| Sample Name | L* | a* | b* | C* | hab |
|---|---|---|---|---|---|
| Day0 | 54.49 ± 0.00 | 32.04 ± 0.00 | 14.65 ± 0.00 | 35.23 ± 0.00 | 2.03 ± 0.00 |
| O-270 | 51.17 ± 0.40 | 33.14 ± 0.21 | 16.89 ± 0.02 | 37.20 ± 0.17 | 1.79 ± 0.01 |
| T-270 | 50.45 ± 0.25 | 31.59 ± 0.25 | 28.40 ± 0.26 | 42.48 ± 0.18 | 0.80 ± 0.02 |
| Y-270 | 50.21 ± 0.32 | 35.33 ± 0.03 | 63.19 ± 0.02 | 72.40 ± 0.05 | 0.22 ± 0.01 |
| G-270 | 48.21 ± 0.03 | 38.63 ± 0.02 | 8.16 ± 0.01 | 39.48 ± 0.04 | 4.66 ± 0.01 |

The resulting L*, a*, and b* values were further color-matched using Adobe Photoshop software (Figure 1). It is evident that with increasing storage time, color difference between wines in different containers became more obvious. Color of wine in stainless-steel tanks did not change much and remained a light purplish red. Wine in unglazed pottery altars and oak barrels became brick red at the late aging stage. Wine in glazed pottery altars became distinctly brick red at the mid-stage, with a more pronounced yellow tint than wine aged in unglazed pottery altars. Yellow hue deepened further in the late stage, becoming a distinct brownish yellow.

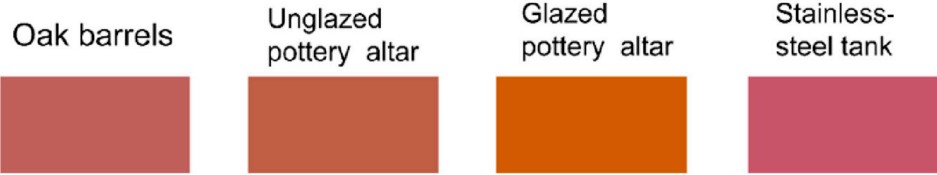

**Figure 1.** Colors of red wine samples in different aging containers on the 270th day.

### 3.2. Qualitative and Quantitative Analysis of Anthocyanins

Based on the UV-Vis spectra, characteristic MS ions, and fragment ions [23–25], a total of 15 free anthocyanins were detected in all samples, including five basic anthocyanins and their acetylated and coumaylated derivatives, as well as pyruvate derivatives and acetaldehyde derivatives of anthocyanins (Table S1). The content of monomer anthocyanins in red wine samples stored in different containers is shown in Table 4. The content of nine types of anthocyanin showed extremely significant differences ($p < 0.01$) among four groups; three types of anthocyanin were significantly different ($p < 0.05$); and the other three types of anthocyanin had no significant differences.

**Table 4.** Types and contents of free anthocyanins in red wine aged in different containers (mg/L).

| Sample Name | Dp | Cy | Pt | Pn | Mv | Dp-acl | Ma-acl-pyr | Pt-acl | Dp-cou | Pn-acl | Ma-acl | Ma-cou-pyu | Pt-cou | Ma-cou | Ma-vin | Total |
|---|---|---|---|---|---|---|---|---|---|---|---|---|---|---|---|---|
| Day0 | 28.98 ± 0.97 | 7.00 ± 0.32 | 40.76 ± 1.44 | 19.93 ± 0.75 | 346.69 ± 9.22 | 15.42 ± 0.80 | 2.07 ± 0.07 | 17.62 ± 0.74 | 1.65 ± 0.12 | 13.42 ± 0.58 | 135.52 ± 4.73 | 2.63 ± 0.39 | 5.01 ± 0.83 | 69.31 ± 3.52 | 1.435 ± 0.07 | 707.43 ± 24.55 |
| O-270 | 2.89 ± 0.16 | 3.15 ± 0.28 | 6.72 ± 0.72 | 2.89 ± 0.05 | 98.42 ± 1.46 | 3.89 ± 0.10 | - | 7.05 ± 0.21 | - | 4.06 ± 0.30 | 49.49 ± 2.16 | 6.3 ± 0.49 | 0.79 ± 0.09 | 23.81 ± 1.05 | - | 209.46 ± 7.07 |
| T-270 | - | 2.98 ± 0.38 | 22.86 ± 0.93 | 2.89 ± 0.04 | 85.33 ± 2.32 | 3.55 ± 0.14 | - | 5.22 ± 0.03 | - | - | 66.62 ± 2.61 | 5.25 ± 0.11 | - | 32.99 ± 0.74 | - | 227.68 ± 7.30 |
| Y-270 | - | 3.33 ± 0.09 | 2.15 ± 0.13 | - | 24.67 ± 1.88 | 1.03 ± 0.03 | - | - | - | - | 17.54 ± 0.56 | 6.83 ± 0.79 | - | 19.18 ± 1.01 | - | 74.71 ± 4.49 |
| G-270 | 2.88 ± 0.06 | 3.50 ± 0.10 | 10.54 ± 0.45 | 2.56 ± 0.02 | 93.97 ± 3.82 | 7.05 ± 0.66 | - | 5.65 ± 0.61 | - | 5.89 ± 0.19 | 59.80 ± 2.26 | 5.60 ± 0.53 | 1.30 ± 0.09 | 23.20 ± 1.32 | - | 221.92 ± 10.11 |
| Significant difference | ** | * | ** | ** | ** | ** | ns | ** | ns | ** | ** | * | * | ** | ns | ** |

The abbreviations and full names of the compounds in Table 4 are as follows: Delphinidin3-O-glucoside (Dp), Cyanidin3-O-glucoside (Cy), Petunidin3-O-glucoside (Pt), Peonidin3-O-glucoside (Pn), Malvidin3-O-glucoside (Mv), Delphinidin-3-O-(6-O-acetyl)-glucoside (Dp-acl), Malvidin-3-O-(6-O-acetyl)-glucoside-pyruvicacid (Ma-acl-pyr), Petunidin-3-O-(6-O-acetyl)-glucoside (Pt-acl), Dephinidin-3-O-(6-O-coumaryl)-glucoside (Dp-cou), Peonidin-3-O-(6-O-acetyl)-glucoside (Pn-acl), Malvidin-3-O-(6-O-acetyl)-glucoside (Ma-acl), Malvidin-3-O-(6-O-coumaryl)-glucoside-pyruvicacid (Ma-cou-pyu), Petunidin-3-O-(trans-6-O-coumaryl)-glucoside (Pt-cou), Malvidin-3-O-(trans-6-O-coumaryl)-glucoside (Ma-cou), Malvidin-3-O-glucoside-4-vinylphenol (Ma-vin). "-" represents "not detected", "ns" represents "not significant", "*"represents "$p < 0.05$", and "**"represents "$p < 0.01$". Statistical analysis was performed by comparing free anthocyanins in red wine aged in four containers.

Among anthocyanins, Malvidin (Ma) occupied the highest content, which was consistent with anthocyanin composition of wines made from Cabernet Sauvignon grapes, as previously reported [26]. Although content of free anthocyanins in red wines aged in the four storage containers all showed a downward trend, the degree of decrease was significantly different. Wine aged in unglazed pottery altars showed the smallest decrease in total anthocyanin content (a reduction of 67.8%), followed by wine aged in stainless-steel tanks (68.6%), oak barrels (70.4%), and glazed pottery altars (89.4%). For some anthocyanins, decline varied differently between different storage containers. For example, Pt content decreased by 43.9%, 74.2%, 83.5%, and 94.7% after aging for 270 days in unglazed pottery altars, stainless-steel tanks, oak barrels, and glazed pottery altars, respectively. In particular, after aging, decrease of Cy content in wine in the oak barrel group and unglazed pottery altar group reached 55.00% and 57.50%, which were higher than that in the stainless-steel

tank group and glazed pottery altar group (52.5% and 50.00%, respectively). This might have been due to better air permeability of oak barrels and unglazed pottery altars, which formed a micro-oxygen environment during the aging process, promoting the transformation of Cy from a free state to a bound state [27,28]. Thus, content of Cy in oak barrels and unglazed pottery altars decreased obviously after aging. By day 270, the number of free anthocyanins had also decreased, with the greatest reduction observed in wines aged in the glazed pottery altar, where only seven free anthocyanins were detected. The other three aging methods also decreased the number of free anthocyanins, but to a lesser extent in wines aged in oak barrels and stainless-steel tanks (12 types each).

In general, change in free anthocyanin quantity was consistent with change in total anthocyanin content, both of which showed a decreasing trend with age. After 270 days of aging, the most types of free anthocyanin (12 types) were detected in wine aged in oak barrels and stainless-steel tanks, while total amount of residual free anthocyanins was the highest in wine aged in unglazed pottery altars (227.68 mg/L).

### 3.3. Qualitative and Quantitative Analysis of Nonanthocyanidins

Nonanthocyanidins in red wine aged in different aging containers were qualitatively and quantitatively detected. A total of 13 nonanthocyanins were detected (Table S2). The content of those 13 monomeric phenols after 270 days of aging in different containers were compared. The results are shown in Table 5.

**Table 5.** Qualitative analysis of nonanthocyanidins in red wine samples (mg/L).

| Monomer Phenolic Compounds | Base Wine | Unglazed Pottery Altar | Glazed Pottery Altar | Stainless-Steel Tank | Oak Barrel | Significant Difference |
|---|---|---|---|---|---|---|
| CAT | 117.75 ± 0.65 | 80.38 ± 1.88 | 49.02 ± 1.61 | 83.54 ± 1.25 | 66.61 ± 1.34 | ** |
| Salicylic acid | 4.27 ± 0.43 | 5.38 ± 0.05 | 5.64 ± 0.10 | 5.42 ± 0.04 | 5.50 ± 0.04 | * |
| Rutin | 1.07 ± 0.25 | 0.48 ± 0.11 | 0.45 ± 0.11 | 0.49 ± 0.06 | 0.47 ± 0.07 | ** |
| Gallic acid | 27.20 ± 3.90 | 93.95 ± 14.15 | 29.68 ± 10.64 | 31.81 ± 4.60 | 103.52 ± 14.98 | ** |
| Protocatechuic acid | 1.42 ± 0.10 | 1.41 ± 0.16 | 1.48 ± 0.17 | 1.40 ± 0.12 | 1.40 ± 0.16 | ns |
| 4-Hydroxybenzoic acid | 0.72 ± 0.20 | 2.17 ± 0.23 | 3.03 ± 0.30 | 1.08 ± 0.20 | 1.53 ± 0.22 | ** |
| Vanillic acid | 7.06 ± 0.22 | 12.68 ± 0.69 | 18.12 ± 0.94 | 8.41 ± 0.39 | 13.23 ± 0.94 | ** |
| Syringic acid | 5.78 ± 1.57 | 25.35 ± 5.08 | 44.56 ± 7.27 | 11.19 ± 3.23 | 21.65 ± 5.76 | ** |
| EC | 49.05 ± 1.75 | 25.06 ± 0.21 | 14.16 ± 0.42 | 32.85 ± 0.54 | 22.57 ± 0.92 | ** |
| EGC | 8.39 ± 0.10 | 5.49 ± 0.15 | 3.50 ± 0.05 | 6.87 ± 0.05 | 4.98 ± 0.15 | ** |
| 2,5-Dihydroxybenzoic acid | 1.27 ± 0.15 | 1.80 ± 0.12 | 1.01 ± 0.18 | 2.21 ± 0.05 | 2.32 ± 0.09 | ** |
| Caffeic acid | 13.96 ± 0.01 | 33.90 ± 1.50 | 32.97 ± 1.80 | 27.06 ± 0.49 | 28.81 ± 0.61 | ** |
| EGCG | 0.83 ± 0.01 | 0.85 ± 0.01 | 0.84 ± 0.01 | 0.90 ± 0.05 | 0.85 ± 0.01 | ns |
| Total | 238.74 ± 9.22 | 288.88 ± 24.33 | 204.43 ± 23.59 | 213.23 ± 10.95 | 273.42 ± 25.27 | ns |

The abbreviations and full names of some compounds in Table 5 are as follows: Catechin Hydrate (CAT), L-Epicatechin (EC), Epigallocatechin (EGC), Epigallocatechin gallate (EGCG). "ns" represents "not significant", "*" represents "$p < 0.05$", and "**" represents "$p < 0.01$". Statistical analysis was performed by comparing nonanthocyanidins in red wine aged in four containers.

According to the above results, it can be seen that content of ten nonanthocyanins was extremely significantly different among the four groups ($p < 0.01$), one type of nonanthocyanin was significantly different ($p < 0.05$), and difference was not significant for the other two nonanthocyanins. After 270 days of aging, the unglazed pottery altar group had the highest total nonanthocyanidin content (288.88 mg/L). Before aging, the most abundant nonanthocyanidin in red wine was CAT, followed by EC, gallic acid, caffeic acid, EGC, vanillic acid, syringic acid, salicylic acid, 2,5-dihydroxybenzoic acid, and rutin, while contents of EGCG, 4-hydroxybenzoic acid, and protocatechuic acid were low.

In terms of total phenolic content, after 270 days of aging, that of the unglazed pottery altar group and oak barrel group was significantly higher than that of the glazed pottery altar group and stainless-steel tank group. The reason might have been that glazed pottery altars and stainless-steel tanks contain more metal ions. These metal ions gradually enter the wine body during aging and react with polyphenols in the wine body to accelerate its oxidation [15,29]. As a result, total phenolic content in the glazed pottery altar group and stainless-steel tank group after aging was lower than that in the unglazed pottery altar group and oak barrel group.

After 270 days of aging, contents of four types of nonanthocyanidins were generally lower than those before aging, and contents of nine types of nonanthocyanidins were higher than those before aging. After 270 days of aging, content of CAT was 80.38 mg/L in wine aged in unglazed pottery altars, 49.02 mg/L in wine aged in glazed pottery altars, 83.54 mg/L in wine aged in stainless-steel tanks, and 66.61 mg/L in wine aged in oak barrels, compared to 117.75 mg/L before aging. This means that CAT content decreased by 31.7%, 58.4%, 29.1%, and 43.4%, respectively; content of EC was decreased by 48.9%, 71.1%, 33.0%, and 54.0%, respectively, compared to the original value; and content of EGC was decreased by 34.6%, 58.3%, 18.1%, and 40.6%, respectively. Based on the data above, the loss of those three nonanthocyanidins in unglazed pottery altars was lower than that in oak barrels. The contents of those three nonanthocyanidins decreased most obviously in glazed pottery altars, possibly because the metal elements of glazed pottery altars could also catalyze redox reactions in wine [30], helping with the degradation of CAT, EC, and EGC.

Among the nine nonanthocyanins with increased content, salicylic acid, gallic acid, syringic acid, and caffeic acid were all consistent with the increasing trend of monomeric phenols reported in the relevant literature [31,32]. For gallic acid, as an example, content in unglazed pottery altars, glazed pottery altars, stainless-steel tanks, and oak barrels increased by 245.4%, 9.1%, 16.9%, and 280.6%, respectively. This indicated that unglazed pottery altars and oak barrels had a strong effect on the accumulation of gallic acid during the aging process of red wine, probably due to substances such as eugenol and hydrolyzed tannins contained in oak barrels gradually dissolving into the wine during the aging process. As a result, nonanthocyanidin content after aging was increased from that before aging [33]. In contrast, each glazed pottery altar was covered with glaze, which prevented contact between the wine and the surface of the container, resulting in little change in gallic acid content. In addition, studies have shown that during the aging process, content of gallic acid also increases significantly after wine undergoes a browning reaction [14,15,32,34]. This may have been one of the main reasons for the increase in gallic acid content after aging in unglazed pottery altars. The contents of five nonanthocyanins, i.e., syringic acid, salicylic acid, protocatechuic acid, vanillic acid, and 4-hydroxybenzoic acid, all increased the most during use of the glazed pottery altar.

### 3.4. Analysis of Antioxidant Activity

Previously, we examined changes in the contents of phenolics in red wine upon aging. Changes in phenolic substances can also cause changes in antioxidant activity [35]. Therefore, we tested the effect of different containers on antioxidant activity by measuring DPPH scavenging ability, ABTS scavenging ability, and FRAP scavenging ability of wine samples. Specific antioxidant activity test results are shown in Figure 2.

In this study, DPPH scavenging abilities of wine samples aged in oak barrels, glazed pottery altars, unglazed pottery altars, and stainless-steel tanks as aging containers were 82.1%, 78.7%, 88.3%, and 79.4%, respectively. Compared to that of base wine, DPPH scavenging ability decreased by 15.2% in oak barrels, 18.7% in glazed pottery altars, 8.8% in unglazed pottery altars, and 18.0% in stainless-steel tanks.

Total ABTS antioxidant activity of wine samples aged in oak barrels, glazed pottery altars, unglazed pottery altars, and stainless-steel tanks was 1.431 mM, 1.395 mM, 1.448 mM, and 1.430 mM, respectively. Compared to that of base wine, ABTS antioxidant activity decreased by 1.7% in oak barrels, 4.2% in glazed pottery altars, 0.5% in unglazed pottery altars, and 1.8% in stainless-steel tanks.

FRAP capacities of wine samples aged in oak barrels, glazed pottery altars, unglazed pottery altars, and stainless-steel tanks were 2.009 mM, 1.625 mM, 2.069 mM, and 1.997 mM, respectively. Compared to that of base wine, FRAP capacity decreased by 19.5% in oak barrels, 34.9% in glazed pottery altars, 17.1% in unglazed pottery altars, and 20.0% in stainless-steel tanks.

In general, after 270 days of aging, the order of antioxidant activity of wine aged in the four containers was as follows: unglazed pottery altar > oak barrel > stainless-steel tank > glazed pottery altar.

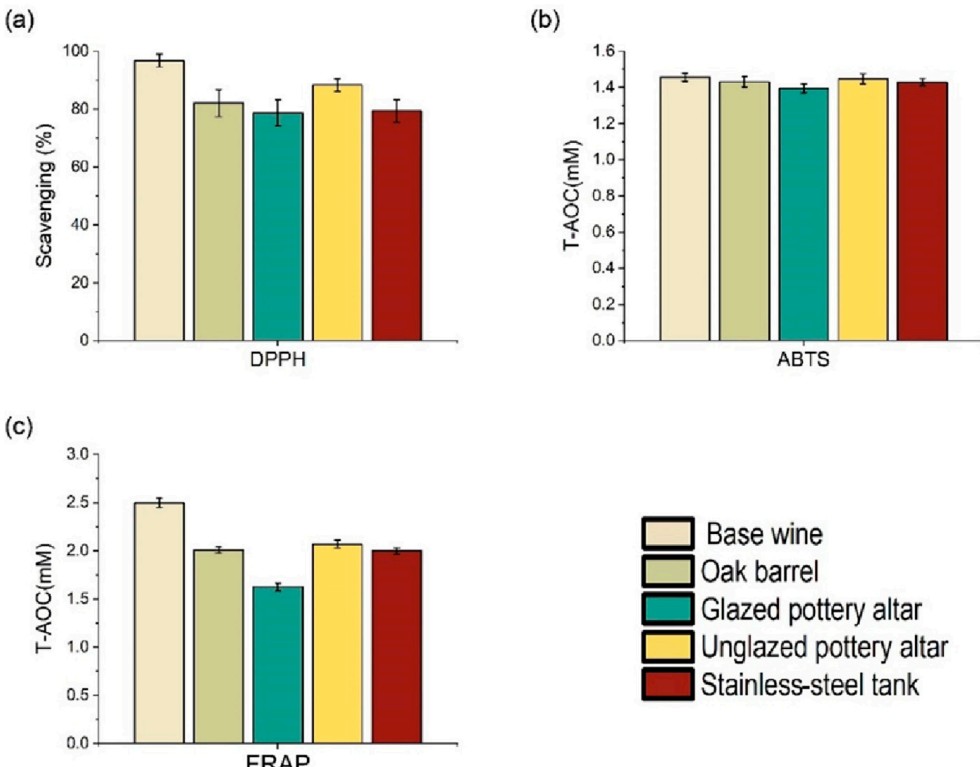

**Figure 2.** Antioxidant activity test results. (**a**) DPPH scavenging ability of red wine aged in different containers. (**b**) ABTS total antioxidant activity of red wine aged in different containers. (**c**) FRAP scavenging ability of red wine aged in different containers.

## 4. Conclusions

To sum up, in this study, oak barrels, glazed pottery altars, unglazed pottery altars, and stainless-steel tanks were selected as aging containers, and changes in color and antioxidant components of red wine after 270 days of aging were analyzed. It is evident that with increased storage time, color difference between wines in different containers became more obvious, while color of wine in stainless-steel tanks did not change much and remained a light purplish red. Total content of residual anthocyanins in red wine aged in unglazed pottery altars was the highest among the four different containers, followed by that of stainless-steel tanks. In addition, better air permeability of unglazed pottery altars and oak barrels formed a micro-oxygen environment during the aging process. Total content of residual nonanthocyanidins in red wine aged in unglazed pottery altars and oak barrels for 270 days was higher than that of red wine aged in stainless-steel tanks and glazed pottery altars under the same experimental conditions. At the same time, DPPH, ABTS, and FRAP scavenging abilities of wine aged in unglazed pottery altars and oak barrels were also stronger than those of wine aged in stainless-steel tanks and glazed pottery altars. Total content of residual nonanthocyanidins and antioxidant ability of wine aged in unglazed pottery altars were higher than those of red wine aged in oak barrels. This shows that red wine can retain most of its antioxidant components when aging in unglazed pottery altars.

Therefore, as one of the oldest wine storage containers in history, the pottery altar is expected to receive more attention from researchers in the future. Through further study of the mechanisms underlying effects of aging in pottery altars on phenolic compounds in wine, unique advantages of the pottery altar as an aging vessel can be discovered. This would make the pottery altar a good choice for wine aging.

**Supplementary Materials:** The following supporting information can be downloaded at https://www.mdpi.com/article/10.3390/fermentation8110592/s1: Table S1: Qualitative analysis of anthocyanins in red wine samples; Table S2: Qualitative analysis of nonanthocyanins in red wine samples.

**Author Contributions:** Conceptualization, C.W. (Chao Wang) and K.T.; methodology, K.T.; software, C.W. (Chenhui Wang); validation, C.W. (Chao Wang), C.W. (Chenhui Wang), Z.R., J.C. and K.T.; formal analysis, C.W. (Chao Wang); investigation, K.T.; resources, Z.R. and J.C.; data curation, C.W. (Chao Wang); writing—original draft preparation, C.W. (Chao Wang); writing—review and editing, K.T.; visualization, C.W. (Chenhui Wang); supervision, Z.R. and J.C.; project administration, Z.R.; funding acquisition, J.C. All authors have read and agreed to the published version of the manuscript.

**Funding:** This research was funded by Development of Oak Barrel Substitute Material Equipment during Wine Aging 2022BBF01003.

**Institutional Review Board Statement:** Not applicable.

**Informed Consent Statement:** Not applicable.

**Data Availability Statement:** Data is contained within the article or Supplementary Material. The data presented in this study are available in Tables S1 and S2.

**Conflicts of Interest:** The authors declare no conflict of interest.

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
