# Peer review of "Effects of Different Aging Methods on the Phenolic Compounds and Antioxidant Activity of Red Wine"

_fermentation, doi:10.3390/fermentation8110592_

Round 1

Reviewer 1 Report

The authors do not specify that the tests were repeated, which makes the results not sufficiently robust for publication.  Moreover, they talk about concentrations of phenolic compounds, but do not specify how the identification and quantification of these compounds was carried out. Furthermore, they say that they have carried out a statistical treatment of the data, but this is not reflected at any point throughout the document. 

On the other hand, in the results section, the authors only limit themselves to describing their tables, but the discussion is very poor. Please expand on this. 

The manuscript needs to greatly improve the discussion, add a statistical treatment to support the data and verify that replicate trials have been carried out. Otherwise, it could not be published. 

Author Response

Thank you very much for sparing the time to read and revise my paper. Thank you for your valuable advice. You have made comprehensive corrections to my paper's structure, content, research methods, and results. It has played a very important role in improving the quality of my paper.

I carefully read the reviewer's comments and revised the paper according to the suggestions, as follows:

  1. Q: Remove please.

A: We have removed the superfluous dot.

  1. Q: Were the trials repeated? If not, the results are not robust and therefore could not be published.

A: All the trials are repeated three times in 2.2. Samples.

  1. Q: Why this sampling date? Please justify.

A: In this study, the main concerns were color and phenolic compounds. The color changes of wines stored in glazed pottery altar and Unglazed pottery altar were very obvious after 270 days. Especially the glazed pottery altar had completely changed to brownish yellow. Therefore, the aging was stopped at 270 days, and all non-volatile substances were analyzed at the same time.

  1. Q: Please add the first author.

A: We have checked that all authors are complete in Reference 18.

  1. Q: What anthocyanin? Please specify the specific anthocyanin.

A: We have specified the anthocyanin as Malvidin-3-O-glucoside. Please see 2.4. Detection of Anthocyanins in Red Wine.

  1. Q: Table 4 refers to different phenolic compounds and their concentration in wine. How have you identified these compounds, do you have calibration lines for each of them, how have you done this quantification?

A: Yes, we have identified these compounds. We also have calibration lines for each of them. We have added table 2 Standard curves of nonanthocyanidins phenolic compounds in 2.6. Detection of Nonanthocyanidins in Red Wine.

  1. Q: Please move this to materials and methods.

A: We have moved this to materials and methods 2.6. Detection of Nonanthocyanidins in Red Wine.

  1. Q: If the quantification has been done considering only one anthocyanin, this should be specified in the table.

A: Since commercial anthocyanins are very few, we used the standard curve of Malvidin-3-O-glucoside for quantification as others usually do

  1. Q: In section 2.4. the authors say that they have carried out three repetitions of the measurements, can they add the standard deviations and a statistical treatment that allows us to see the significant differences?

A: We have added standard deviations and a statistical treatment in Table 4.

  1. Q: Specify in the table footnote what each of the abbreviations means.

A: We have explained the meaning of each abbreviation in the table footnote.

  1. Q: Please include statistical treatment and meaning of abbreviations.

A: We have added a statistical treatment in table 5 and explained the meaning of each abbreviation in the table footnote.

  1. On the other hand, in the results section, the authors only limit themselves to describing their tables, but the discussion is very poor. Please expand on this.

A: We have added corresponding discussion in results and the conclusions.

Finally, thank you again for your guidance and for reviewing and revising my paper again. I hope that I can complete this excellent paper under your guidance, and sincerely hope that my paper can be published in this journal.

Reviewer 2 Report

The object of the study - aging methods/storage of red wine - is interesting and in compliance with contemporary trends.

Overall, the manuscript is well written and informative, it fits into the scope of the Fermentation journal.

Author Response

Thank you very much for sparing the time to read and revise my paper. Thank you for your valuable advice. You have made comprehensive corrections to my paper's structure, content, research methods, and results. It has played a very important role in improving the quality of my paper. I hope that I can complete this excellent paper under your guidance, and sincerely hope that my paper can be published in this journal.

Reviewer 3 Report

The article covers an interesting topic regarding the effects of the four different aging methods of Cabernet Savignon vine on the phenolic content and antioxidant activity of the red wine under study. The article is primarily of practical value, although the authors have performed many analyzes with advanced and complex analytical techniques. The article is a continuation of the earlier works of the authors, which are cited in the references. I rate the scientific value of the article rather low. Despite a large number of complicated measurements, the authors did not undertake an in-depth interpretation of these results and did not answer the question of why wine stored in unglazed ceramic containers retains the highest concentration of polyphenols. Unfortunately, such a wine loses its original color, which is not without significance when assessing the quality of the wine. Here, too, the authors did not explain why the color of the wine changes so significantly after 270 days of aging in different containers.

The advantage of the article is the performance of many studies and analyzes, however, the interpretation of their results and the conclusions drawn are too poor. I would expect the authors to deepen their conclusions about the chemical or physicochemical analysis of the results of the research.

In point 2.1. "Chemicals" , the authors do not mention hydrogen peroxide solutions. What was the starting concentration of this solution? This should be completed. The authors do not provide a list of symbols and acronyms. For a better understanding of the content of the article, the authors should complete this.

Author Response

Thank you very much for sparing time to read and revise my paper. Thank you for your valuable advice. You have made comprehensive corrections to my paper's structure, content, research methods, and results. It has played a very important role in improving the quality of my paper.

I carefully read the reviewer's comments and revised the paper according to the suggestions, as follows:

  1. Q: Cancel BOLD

A: We have fixed the problem of capitalizing words.

  1. Q: Where is explanation of this acronime?(ABTS)

A: We have added a description of the principles of ABTS Free Radical Scavenging Ability in 2.7. Determination of ABTS Free Radical Scavenging Ability.

  1. Q:Where is explanation of this acronime?(FRAP)

A: We have added a description of the principles of FRAP Reducing Iron Ability in 2.8. Determination of FRAP Reducing Iron Ability.

  1. Q: What was the initial (before dilution) concentration of hydrogen peroxide solution?

A: We are sorry that we have checked the relevant literatures on the use of the same kind of kit, and there is no mention of the concentration of hydrogen peroxide solution. In addition, due to commercial competition, the concentration of hydrogen peroxide solution is not stated in the kit's instruction manual.

  1. Q: The advantage of the article is the performance of many studies and analyzes, however, the interpretation of their results and the conclusions drawn are too poor. I would expect the authors to deepen their conclusions about the chemical or physicochemical analysis of the results of the research.

A: We have added corresponding discussion in results and the conclusions.

Finally, thank you again for your guidance and for reviewing and revising my paper again. I hope that I can complete this excellent paper under your guidance, and sincerely hope that my paper can be published in this journal.

Round 2

Reviewer 1 Report

It´s ok. 

Author Response

Thank you for your suggestions. A colleague fluent in English writing has checked the manuscript again and revised it. In fact, we have improved English through professional services before submission. We can make further changes if necessary.
